# Synthetic Data Generation for AI-Based Quality Inspection of Laser Welds in Lithium-Ion Batteries

**DOI:** 10.3390/s25237301

**Published:** 2025-12-01

**Authors:** Jonathan Zender, Stefan Maier, Alois Herkommer, Michael Layh

**Affiliations:** 1Institute for Machine Vision, University of Applied Science Kempten, Bahnhofstr 61, 87435 Kempten, Germany; 2Institute of Applied Optics, Pfaffenwaldring 9, 70569 Stuttgart, Germany; 3Swoboda Wiggensbach KG, Max-Swoboda-Straße 1, 87487 Wiggensbach, Germany

**Keywords:** weld inspection, synthetic data, automated labeling, AI-based quality inspection, physically based rendering

## Abstract

Manufacturing companies are increasingly confronted with critical challenges such as a shortage of skilled labor, rising production costs, and ever-stricter quality requirements. These challenges become particularly acute when defect types exhibit high visual variance, making consistent and accurate inspection difficult. Traditionally, visual inspection of high variance errors is performed manually by human operators—a process that is both costly and prone to errors. Consequently, there is a growing interest in replacing human inspection with AI-based visual quality control systems. However, the adoption of such systems is often hindered by limited access to training data, labor-intensive labeling processes, or the absence of real production data during early development stages. To address these challenges, this paper presents a methodology for training AI models using synthetically generated image data. The synthetic images are created using Physically Based Rendering, which enables precise control over rendering parameters and facilitates automated labeling. This approach allows for a systematic analysis of parameter importance and bypasses the need for large real training datasets. As a case study, the focus is on the inspection of laser welds in battery connectors for fully electric vehicles—a particularly demanding application due to the criticality of each weld. The results demonstrates the effectiveness of synthetic data in training robust AI models, thereby providing a scalable and efficient alternative to traditional data acquisition and labeling methods. The trained binary classifier reaches a precision of 0.94 with a recall of 0.98 solely trained on synthetic data and tested on real image data.

## 1. Introduction

In modern manufacturing, companies face increasing challenges, including a shortage of skilled labor, rising production costs, and higher demands for product quality. These issues are especially critical in quality assurance processes, where strict quality requirements must be met despite considerable variability in the appearance of defects [1,2,3,4,5]. This variability complicates reliable defect detection using image processing. Traditionally, when rule based image processing fails, quality control has relied on manual visual inspection, a method that is not only expensive and time-consuming but also inherently prone to human error [6,7]. To overcome these limitations, industry has increasingly turned to AI-based visual inspection systems. However, the successful deployment of such systems is often hampered by several practical obstacles: the availability of high-quality training data is limited, data labeling is labor-intensive and costly and, in some cases, the production process has not yet begun—making it impossible to collect real data for training purposes [8,9]. To address these challenges, this work explores the use of Physically Based Rendering (PBR) for generating synthetic training data.

As a case study, the focus of this work is on the optical inspection of laser welds in battery connectors for fully electric vehicles; see Figure 1. Given the widespread use of laser welding in industries such as aerospace, automotive, and electronics, the proposed methodology is designed to be transferable to other application scenarios [10,11]. The battery connector welds are subject to particularly stringent quality requirements, as a single defective connector can render an entire battery unit unusable. Compounding the difficulty, both the welds and their defects exhibit high visual variance, making them a challenging benchmark for AI-driven inspection.

This paper shows a methodology for generating and using synthetic image data to train AI models for visual inspection tasks without the need for real image data. Different from other approaches, e.g., Generative Adversarial Networks (GANs) that need real image data as an input to generate synthetic training data, the approach described in this work is based on a 3D-model of laser welds, which is used for image rendering. This work demonstrates how this approach enables reliable defect detection and classification while avoiding the costly and time-consuming process of real image data collection and manual labeling. Additionally this approach also allows for an effective design of optical inspections systems before start of production.

The goal is to investigate whether synthetic datasets can be generated without real image data to train neural networks for quality inspection prior to start of production. To enable broad industrial applicability across different types of laser welds, the analysis examines which scene parameters require the highest modeling accuracy.

## 2. Related Work

Nasir Ud Din et al. [12] conducted a comprehensive study addressing defect detection in battery laser welds using deep learning. They trained a convolutional neural network on a dataset comprising 3736 real images, of which 3270 depicted various defect types. The authors defined eight distinct defect classes, including typical anomalies such as cracks, pores, and spatter irregularities. Their model achieved high performance, reporting both recall and precision scores of 0.94, demonstrating the feasibility of deep learning for reliable and scalable quality inspection in battery manufacturing processes. Wang et al. [13] conducted a study using a dataset of 1268 weld images, including 208 defective samples, to develop a deep learning-based inspection method for laser-welded power battery components. They proposed a two-branch neural network in which one branch performs segmentation of the weld from the full image, while the other branch classifies the weld as either OK or NOK. The binary classification branch achieved a maximum accuracy of 0.9696. Xin et al. [14] present a vision-based optical inspection method for detecting welding defects in bright stainless-steel thin plates. They note that traditional 3D measurement techniques can struggle with strong specular reflections on highly reflective metal surfaces, which reduce measurement accuracy, although 3D inspection remains necessary in some cases, such as detecting misalignment defects. To address the reflection issue, the authors focus on 2D imaging and introduce a multi-color illumination strategy that minimizes specular effects and improves image quality. Based on 553 real weld images, they train several U-Net-based deep learning models for pixel-level defect segmentation. Among these, the Res-U-Net architecture achieves the best performance, with an F1-score of 0.8841. Yang et al. [15] proposed an enhanced version of the YOLOv5 object detection architecture to improve the identification of laser welding defects in lithium battery poles. Their study utilized a dataset comprising 8439 labeled real images of welding defects, focusing on the two most frequently occurring defect types: holes and bulges. Specifically, the training set included 5231 images depicting welding holes and 3208 images showing bulges. When trained using the standard YOLOv5 model, the network achieved a mAP@0.5 of 95.9%. With their improved YOLOv5 variant, which incorporated architectural refinements tailored to defect characteristics, they further increased the mAP@0.5 to 97%, showing that the YOLO Object Detector classes show good performance on the specific task. The above works demonstrate that AI-based quality inspection performs well on welds when sufficient real image data is available. However, they do not employ synthetic image data for this specific task. Related research by Boikov et al. [16] applied PBR for defect detection and classification on steel plates, illustrating the potential of synthetic data in similar industrial inspection tasks. They defined four defect categories: scratches, surface cracks, network cracks, and caverns. To model these defect types, they primarily utilized Perlin noise to generate defect masks, which allowed for significant variation in the appearance of the defects and facilitated automated labeling through the generated masks. They trained two neural networks on the synthetic image datasets: the first, an object detector, achieved a Dice Score of 0.62, while the second, a classifier for defect categories, reached an accuracy of 0.81. Fulir et al. [17] focused on segmenting defects on complex metallic surfaces—particularly milled automotive components such as clutches—using synthetic data. They created a dataset by modeling synthetic surface defects including scratches and dents on 3D models of milled surfaces. Various segmentation network architectures, such as U-Net and DeepLab variants, were trained exclusively on this synthetic data and evaluated on real images. Despite the domain gap, their approach achieved a maximum $F_{1}$ score of 40%, indicating a promising direction for transferring synthetic learning to real quality assurance tasks. The study highlights the challenges of domain adaptation but also the potential for synthetic data to serve as a valuable resource in defect segmentation when annotated real data is limited. The above works showed that synthetic data can efficiently train neural networks that perform well on real image data.

## 3. Methods

### 3.1. Real Image Recording Setup

The experimental setup is configured as shown in Figure 1a. The primary illumination is provided by an area light source featuring a through-hole, through which the lens is positioned to capture images of the welds. The camera captures an entire circular weld as shown in Figure 1b. Given a working distance of 216 mm, the Edmund Optics 56-872 lens with a focal length of 35 mm and the Basler a2A4200-12gmPRO camera with a resolution of 4200 × 2160 pixels, resulting in a spatial resolution of approximately 13 μm. To facilitate the processing for neural networks, the circular welds, as seen in Figure 1b, are first detected and then unwrapped after image acquisition, as seen in Figure 2a. The unwrapped weld is subsequently divided into 21 cropped segments of 128 × 128 pixels, as shown in Figure 2b–e. These image crops are then used as input for the neural networks.

### 3.2. Real Dataset

The present study accompanies a real industrial ramp up process as it progresses towards production. Consequently, the collection of a real dataset is a highly challenging endeavor, which is only feasible to a limited extent. The primary reason for this is that the product is still in the prototype phase, so no sample parts are yet available. Furthermore, the laser welding process is under optimization to ensure operational stability, thereby minimizing the production of defective components. Consequently, the real dataset is relatively small and contains only 50 defective components. As described in Section 3.1, 21 images for classification are generated from a single recording. By applying a sliding crop window with a step size of 10 pixels, 1671 images with defects (NOK) are obtained: 1495 images containing the defect type “hole” and 176 images containing the defect type “missing weld”. Hermens [18] found that a YOLO model trained on only 100 images already achieves strong performance. Therefore, the 176 images available for the “missing weld” class are assumed to be sufficient. To obtain a balanced dataset, the approach proposed by Nasir et al. [12] is applied and the “missing weld” class is upsampled, by duplicating the images, to match the number of samples in the “hole” class. Additionally, the images without defects (OK) are downsampled to the same number (2990). From this balanced dataset of 5980 images, 70% are split into a training set and 15% each into validation and test datasets. The training set is then used to train the benchmark networks on real data. Real image examples are depicted in Figure 2.

### 3.3. Generation of Synthetic Welds

The following section outlines the methodology used to generate synthetic image data of welds. The synthetic weld generation scheme is illustrated in Figure 3. The geometry of the welds, its defects, and material properties of the scene are modeled. To synthesize the welds a single 3D surface measurement of a real weld is used, as shown in Figure 4a. The 3D measurement is taken from an OK part, typically gathered during laboratory tests prior to the start of production. Thus, this approach enables transferability to other laser welds, as the modeling of the synthetic welds is based on measurement data. Another advantage of this method is that, as shown by Xin et al. [14], performing inspections on 3D image data can be beneficial. Since the welds are modeled from 3D measurements, it is also possible to generate synthetic 3D data. The 3D measurement is obtained with a white light interferometer with a lateral resolution of 3 μm and a lateral measurement range of 6 mm × 6 mm. Afterwards, the 2D Fourier transformation is analyzed and calculated with Equation (Equation 1) [19]. The spatial frequencies that have at least 10% amplitude of the highest amplitude of the 2D Fourier transformation are extracted, resulting in a filtered spectrum ($F_{f}$). Based on the filtered spectrum, synthetic welds are generated by adding a random amplitude change of ±15% and a random phase to the spectrum by applying the inverse Fourier transformation as achieved by Equation (Equation 2).(1)$$F\left(u,v\right)=\sum_{x=0}^{M-1}\sum_{y=0}^{N-1}z_{m}\left(x,y\right)\;\cdot \;e^{-2\pi i\left(\frac{ux}{M}+\frac{vy}{N}\right)}$$(2)$$z\left(x,y\right)=\frac{1}{MN}\sum_{u=0}^{M-1}\sum_{v=0}^{N-1}s\left(u,v\right)\;\cdot \;\left|F_{f}\left(u,v\right)\right|\;\cdot \;e^{i\phi (u,v)}\;\cdot \;e^{2\pi i\left(\frac{ux}{M}+\frac{vy}{N}\right)}$$
In Equation (Equation 2) of the weld generation equation, *s* denotes a random amplitude variation of ±15%, corresponding to a range from 0.85 to 1.15. The variable ϕ represents the random phase and $F_{f}$ is the filtered Fourier spectrum. Subsequently, a random yet comparable surface is obtained, arranged according to frequencies and amplitudes, resembling a weld. As depicted in Figure 4a, there is another “scaly” pattern on the real weld. These scales are the result of the laser moving only in one direction during the welding process. Thus, the synthetic surface pattern must also be shifted in the same direction. To simulate the scales, the phase of the dominant amplitude is analyzed row-wise of the weld measurement, as shown in Figure 4b. Here, it is noticeable that the scales can be approximated with a Gaussian curve. Thus, to simulate the movement of the circular laser beam, the pixel rows are shifted along the x-direction. The shift distance depends on the y-position and follows a Gaussian curve. Furthermore, upon analysis of the measurement, it becomes evident that the sides of the weld exhibit a reduced heights in comparison to the center. Consequently, a height reduction procedure is applied to the weld edges by introducing a Gaussian weighting function along the y-direction, while maintaining translational invariance along the x-direction. The resulting profile is approximated by a Gaussian function, as shown in Figure 4c. In this instance, the approximation does not appear to approximate the weld. This is due to an overswing at one side of the measurement in Figure 4c. The rationale behind this phenomenon is that one weld partner, in the measurement, is situated at a higher elevation than the other weld partner. The result of Equation (Equation 2) is shown in Figure 5a while in Figure 5b the shifted surface is depicted. In Figure 5c the surface from Figure 5b with reduced edge height is shown.

### 3.4. Insert Variance into the Weld Images

In order to enhance the variance of the weld images, a range of options are presented. The influences can be divided into two groups based on their evaluation. The primary factor under consideration is the influence of the geometry of the weld. The second factor under consideration is the influence of rendering effects, such as the use of different Bidirectional Scattering Distribution Function (BSDF)’s to generate vertical stripes on the weld partners—originating from the preprocessing of the component—or to simulate soot on the weld. Within Section 8.3, the impact of each influencing factor on test metrics is analyzed.

The following parameters are investigated and explained in the following section, except for the modeled illumination, which is explained in Section 4.2.

Geometric

modeled weld topographyreduced weld edge heightsrandom border contourPerlin noise

Rendering

weld partner surface variancesmodeled soot on weldmodeled soot at edges of weldmodeled illumination

#### 3.4.1. Geometry Based

##### Modeled Weld Topography

Here, the necessity of modeling the weld topography is investigated. Two scenarios are considered: one in which a weld topography is modeled, as described in Section 3.3, and one in which no weld topography is modeled, i.e., the weld is a flat surface. In Figure 8a–c,e,f, a weld topography is modeled, whereas in Figure 8g,h, no weld topography is modeled.

##### Reduced Weld Edge Heights

As demonstrated in Figure 4c, the weld amplitude is higher in the center than at the borders. To simulate the reduced amplitudes between weld partners and weld, the Gaussian weighting model described in Section 3.3 is applied. It reduces the depth difference between the edges of the weld and the weld partners. In Figure 8e, the edges are reduced, whereas in Figure 8f, the edges remain unreduced.

##### Random Border Contour

In the context of the mask-based methodology, explained in Section 3.5, the transition from the weld partners to the weld is always a straight line. This will never be the case on the real weld. Thus, randomly changing border contours are added from weld to weld partners. In Figure 8a–c,e,g, the contour between the weld and the weld partner is randomized, whereas in Figure 8f,h, the edges remain straight.

##### Perlin Noise

As proposed by Boikov et al. [16] and Ebert et al. [20], an additional structural element was incorporated onto the surface, based on Perlin noise. The Perlin noise is employed to simulate the discontinuities of the weld that are an inevitable consequence of the welding process. This is because the Fourier spectrum lacks high-frequency components and cannot accurately represent discontinuities. In Figure 8a–c, Perlin noise is applied to the weld, whereas in Figure 8e–h, no Perlin noise is applied.

#### 3.4.2. BSDF-Based

##### Weld Partner Surface Variance

As shown in the real image in Figure 2b, the weld partners exhibits vertical surface variances that are discernible to the naked eye, which are result of the preprocessing of the component. The purpose of this supplement is to model these material-based surface variances. The BSDF changes are applied to the weld partners during the rendering process. Rendered weld partner surface variances are depicted in Figure 8a–d,f,g.

##### Modeled Soot on Weld

During the welding process, soot and other minor but undesirable contaminants can be produced, which are deposited on the weld. To address this, the BSDF of the weld is subject to random modification, resulting in the occurrence of darker regions in random areas of the weld. It is hypothesized that this will increase the variance of the weld, not only based on the surface normal, but also based on the BSDF, with an example depicted in Figure 8h.

##### Modeled Soot at Edges of Weld

With the occurrence of soot, the edges of the welds often seem darker than the center of the welds, as seen in Figure 2c. To simulate the soot, a less-reflective BSDF is blended at the edges to darken the edges of the weld, as shown in Figure 8f,g.

The combination of the above parameters together with the varying illumination results in 256 distinct synthetic datasets on which neural networks are trained.

### 3.5. Weld Error Generation

The methodology employed in this study introduces errors into the mesh through a modeling process. This approach is based on the use of UV maps specific to the designated mesh. Implementing this method requires modeling the errors as crop masks. The following section describes the methodology used for generating these crop masks. Huang et al. [21], Din et al. [12], Yang et al. [15], and Xin et al. [14] showed that the following simulated weld errors are the most common to occur.

#### 3.5.1. Hole

The holes are generated using deformed ellipses characterized by two major axes. In the subsequent step, a deformation factor is added to the ellipse. The construction formula is demonstrated in Equation (Equation 3). These deformed ellipses are then placed randomly on the crop mask, as seen in Figure 6a.(3)$$\begin{array}{cc} x_{d}\left(x\right) & =a\cos(x)+d\sin(ax)\;a\in \{3,4,5\},b\in \{2,3\}\\ y_{d}\left(x\right) & =b\sin(x)+d\cos(bx)\;x\in [0,2\pi ],d\in [0,1] \end{array}$$

#### 3.5.2. Missing Weld

In order to synthesize the missing weld, it was not necessary to introduce an actual weld. Consequently, the mesh can be divided into two components. Since the separation of the weld partners is not always a horizontal line, as seen in Figure 2d, the separation is achieved through a combination of sine waves with varying phases ($\phi_{i}$), as delineated in Equation (Equation 4). The amplitudes ($A_{i}$) and frequencies ($\nu_{i}$) are randomly sampled from a normal distribution. Once the trajectory of the missingweld is defined, its width is specified by incorporating the adjacent pixels into the mask. A resulting mask is presented in Figure 6b.(4)$$y_{m}\left(x\right)=\sum_{i=1}^{N}A_{i}\;\cdot \;\sin\left(2\pi \nu_{i}x+\phi_{i}\right)$$

## 4. Rendering

The present work employs the PBR technique to generate synthetic image data. PBR enables fine-grained control over image generation parameters, allowing for realistic simulation of various defect types and systematic variation of inspection conditions. Furthermore, in contrast to alternative approaches, such as GANs, PBR facilitates the mathematical traceability of all parameters, thereby enabling a systematic approach to the determination of rendering parameters [22,23,24]. Another major advantage of the described approach is that no real images are required to create the synthetic datasets, whereas GANss rely on real data for training [25,26]. This limits the use of GANs prior to the start of production and restricts the ability to adjust training datasets to new weld types.

For instance, illumination conditions—including intensity, wavelengths, and texture—can be measured or adjusted, and object-scattering properties can be modified by influencing the BSDF of the object. The BSDF has been demonstrated to capture the non-optically resolvable spatial frequencies of a surface, thereby affecting the object’s optical appearance [22,27]. The rendering of a scene utilizing PBR necessitates the implementation of an illumination characterized by defined, emitted wavelengths and a spatial and angular intensity distribution, also referred to as “texture”. Furthermore, the scattering of the surface must be specified in terms of surface normals and optical material properties. In this study, the object is modeled as a mesh representation of individual welds. The surface scattering properties are simulated using a BSDF—specifically a “rough conductor” material—from the utilized Mitsuba3 Physically Based Renderer [22]. The “rough conductor” is employed with the Beckmann distribution and an alpha parameter, which is defined in the next section.

### 4.1. Bsdf Parametrization

The Mitsuba3 rough conductor BSDF is applied to the meshes, where it can be controlled by an alpha value. The alpha value characterizes the angular width of the scattering cone when a ray interacts with the surface [22]. To determine the optimal alpha value for the weld and weld partner, images are rendered using alpha values ranging from 0.02 to 0.5 in increments of 0.02. A grid search is performed, meaning that all possible combinations of alpha values for the weld and the weld partner are rendered. Each combination is rendered five times to mitigate the effect of outliers. Each rendered image is then divided into two regions: the weld and the weld partners. The same segmentation is applied to the real images. To evaluate the similarity between rendered and real images, the chi-square distance is computed using a state-of-the-art technique for assessing image similarity based on histograms [28,29,30]. For each parameter combination, histograms are computed for both regions of the real and rendered images. The alpha values that yield the minimum chi-square distance are identified as the best match to the real material appearance. The relationship between the alpha parameter and the chi-square distance is illustrated in Figure 7.

### 4.2. Illumination Condition

As delineated in Section 3.1, the probe is illuminated by an area emitter. Here, it is analyzed whether it is necessary to simulate the correct illumination conditions. The importance of accurately modeling the radiation characteristics of the area illumination in angular space is investigated. To this end, two illumination conditions are examined. In the first case, an area emitter with the same shape as in the real image setup is used. Due to the implementation of the Mitsuba renderer, the area emitter can only emit Lambertian-distributed light [22]. Since the real area emitter does not emit light with a Lambertian distribution, a second illumination condition is modeled, referred to as the modeledillumination, which features a Gaussian radiation characteristic. This is implemented by arranging an array of 33 spotlights with a Gaussian radiation pattern in angular space and an opening angle corresponding to the real setup, following the same shape as the real area emitter. In summary, two illumination conditions are evaluated: one that emits light with a Lambertian distribution, and the modeledillumination, which emits light with a Gaussian falloff. These two configurations are included in the rendering parameters discussed in Section 8.

### 4.3. Automated Labeling

Since the whole setup is metric and physically correct, pixel-precise labeling can be performed by applying the camera equation from Equation (Equation 5) [31], where u,v stands for the pixels in the rendered image, K stands for the camera intrinsic parameters, and R,t are the camera extrinsic parameters, which are all defined in the render software. X,Y,Z are the vertice positions of the errors. Applying this equation results in pixel-precise labeling, which can be converted to bounding box or classification labeling. The labeling scheme is shown in Figure 3.(5)$$\left[\begin{array}{c} u\\ v\\ 1 \end{array}\right]=K\left[\begin{array}{cc} R & t \end{array}\right]\left[\begin{array}{c} X\\ Y\\ Z\\ 1 \end{array}\right]$$

## 5. Neural Network

As proposed by Yang et al. [15], the YOLO network is employed for image analysis. In this study, the object detector YOLO-NAS-S is used, developed by SuperGradients [32], as the object detection network. It is trained exclusively on a single error class, which can be either a “hole” or a “missing weld”. Since the detector is trained on only one class, the task can be formulated as a binary OK-NOK classification problem. Consequently, the YOLOv11 classified from Ultralytics [33] is investigated as a second model architecture.

## 6. Synthetic Dataset

Each synthetic dataset consists of 5000 rendered images, comprising 2500 OK images and 2500 NOK images. Among the NOK images, 1250 depict hole defects, and the remaining 1250 represent missing weld defects. The images are generated on a Nvidia L40S GPU with 20 cores of AMD EPYC 9534 CPU. The generation of one dataset takes approximately 10 min, while training one YOLOv11 classifier takes around 4 min to train on the used GPU.

## 7. Training

Given that the primary focus of this study is the establishment of a production line, the training of the neural networks is conducted exclusively using synthetic data. Accordingly, for each combination of parameters described in Section 3.4, a corresponding dataset was rendered. In total, 256 datasets were generated, and a separate neural network was trained for each dataset. The trained networks were evaluated and tested on real images. In this way, the principal factors influencing the generation and rendering of the welds can be identified, thereby enabling an assessment of their impact on the neural network test metrics. The models were trained using the Adam optimizer with a learning rate of 0.0005 and a batch size of 30.

## 8. Results

The work accompies a initial phase of an industrial ramp-up process, where complete visual inspection is mandatory, the primary objective is to optimize the networks for recall. Recall reflects the proportion of defects not identified by the neural network [34,35]. Recall (*R*) is calculated following Equation (Equation 6) [36]. In contrast, precision quantifies the proportion of misclassified parts, and thus directly affects production costs [37]. Precision (*P*) is computed as defined in Equation (Equation 7) [36]. To balance these two measures, the $F_{2}$ score as a combined metric is applied. Since a single defective part can render an entire component unusable, recall is weighted more heavily in this score. The $F_{2}$ score, a variant of the $F_{1}$ score with recall given twice the weight, is calculated as defined in Equation (Equation 8) [36]. Based on this metric, the best-performing networks are selected from the 256 distinct models.(6)$$R=\frac{\mathrm{TP}}{\mathrm{TP}+\mathrm{FN}}$$(7)$$P=\frac{\mathrm{TP}}{\mathrm{TP}+\mathrm{FP}}$$(8)$$F_{2}=\frac{5\;\cdot \;P\;\cdot \;R}{4\;\cdot \;P+R}$$

### 8.1. Desired Test Metrics

In this production case, a precision of 0.85 and a recall of 0.95 is required, which surpasses the recall performance of human visual inspection. For instance, Caballero-Ramirez et al. [6] reported a human inspection recall of 0.924 for floral wreaths, while See [7] found that human visual inspection recall typically has a range of 0.7–0.8, indicating significantly lower performance.

### 8.2. Test Metrics of Trained Networks

In Table 1, the test metrics of the models that achieve the highest $F_{2}$ scores are presented. It is important to note that models 1 and 3 are trained on synthetic data, while models 2 and 4 are trained on real data. All models are tested on real image data. The object detector trained on synthetic data reaches good results, but it is not yet in the range that is required from the industry, as described in Section 8.1. In comparison, the synthetic trained binary classifier already reaches the desired test metrics on real image data. The best-performing binary classifier was trained on a synthetic image dataset that is generated with a modeled weld, with Perlin noise, with reduced height at weld edges, with random border contour, with modeled soot on weld, and with the modeled illumination but without modeled soot at the edges of the weld, as seen in Table 2. Example images of the best combination are shown in Figure 8a–d.

### 8.3. Sensitivity Study on Modeling and Rendering Parameters

Since the binary classifier achieves high test metrics, the sensitivity study of the dataset generation parameters is conducted only for this model. The study is performed on the dataset that achieved the highest $F_{2}$ score for the binary classifier, as shown in Table 1. As described earlier, 256 distinct models were trained on 256 different synthetic datasets. Each dataset was generated using a unique combination of the parameters described in Section 3.4. The influence of each generation parameter on the test metrics is analyzed. Therefore, first, the distribution of the $F_{2}$ scores across all 256 models is examined, as seen in Figure 9. It is evident that some models perform very well, while others perform poorly. This is due to the wide range of generation parameters used in the experiments, including configurations in which no weld is modeled at all. To identify which generation parameters has the greatest influence on the test metrics, the dataset with the highest $F_{2}$ score is selected as baseline. This dataset was generated with all parameters enabled except for the soot simulation at the weld edges.

Using this generation configuration as baseline, a sensitivity is performed by changing only one parameter at a time to measure how much the test metrics decreases. Thus, each new dataset differs from the baseline dataset by only one generation parameter. See Table 2 for the exact parameter configuration of each model. The results are shown in Figure 10. The results clearly indicate that three parameters cause a substantial drop in performance when disabled: modeled weld topography, Perlin noise, and modeled illumination. Disabling either the modeled weld topography or the Perlin noise results in an $F_{2}$ score reduction of nearly 80%, while disabling the modeled illumination reduces the $F_{2}$ score by approximately 40%. All other parameters have only minor effects on the test metrics. These results indicate that, for the generation of synthetic data for weld inspection, the most critical factors are as follows:Accurate modeling of weld topography with high variance;Inclusion of Perlin noise to simulate weld discontinuities;Correct simulation of illumination conditions.

**Table 2 sensors-25-07301-t002:** The best model is trained on synthetic dataset, where all parameters are enabled (•) except modeled soot at edges (◦). The models M1–M8 are trained on datasets where only one parameter is always inverted with respect to the best model. The influence on the test metrics is shown in Figure 10.

	Weld Topography	Reduced Weld Edges	Random Contour	Perlin Noise	Modeled Illumination	Partner Variances	Soot on Weld	Soot at Edges
best model	•	•	•	•	•	•	•	◦
M1	◦	•	•	•	•	•	•	◦
M2	•	◦	•	•	•	•	•	◦
**⋮**	⋮	⋮	⋮	⋮	⋮	⋮	⋮	⋮
M8	•	•	•	•	•	•	•	•

Parameters such as random border contour, reduced weld edges, weld partner surface variance, modeled soot on the weld, and modeled soot at the weld edges have only marginal influence. In the next step, a dataset with only the three most influential parameters enabled and all others disabled is rendered. The results, shown in the second row of Table 3, demonstrate that the test metrics are nearly as good as those obtained with the full parameter set. This confirms that modeling only weld topography, Perlin noise, and illumination is sufficient for generating high-quality synthetic data for weld inspection. Finally, it is investigated whether both geometry-based parameters—modeled weld topography and Perlin noise—must be used simultaneously. Two models are compared: one trained on a dataset with only weld topography and illumination enabled, and another trained on a dataset with only Perlin noise and illumination enabled. The results, presented in row 3 and 4 of Table 3, show that neither model reaches the desired test metrics. This finding confirms that both parameters must be modeled together to achieve optimal performance.

## 9. Conclusions

In this study, it is demonstrated that it is feasible to train neural networks solely on synthetic data and achieve strong performance when testing on real data, even in scenarios involving complex surface structures and high variability in appearance. Notably, the binary OK/NOK classifier achieved excellent test metrics, where the most elaborated model with various Geometric and BSDF-based variances in synthetic data generation parameters reaches a precision of 0.94 and a recall of 0.99, being solely trained on synthetic data and tested on real image data. Furthermore, the impact of different synthetic data generation parameters on Neural Network model performance is systematically analyzed. The results show that the geometric modeling with a high variance of welds has the most significant influence on test metrics. Specifically, increasing the variability in the geometric representation of the welds led to a marked improvement in model performance. Additionally, accurately simulating the illumination conditions was found to be crucial for achieving high test scores. It is shown that only three synthetic data generation parameters need to be modeled to achieve the same precision of 0.94 and recall of 0.98 as with fully elaborated modeling. Interestingly, BSDF-based variances were found to have a comparatively minor effect on prediction outcomes. These findings underscore the importance of prioritizing geometric diversity and correct illumination modeling in synthetic data generation, particularly when targeting industrial inspection tasks involving complex surface geometries.

## 10. Outlook

It is planed whether this approach can be transferred to other laser welds, potentially involving different materials such as copper or different laser parameters. Since most geometric parameters are derived from a single measurement of an actual weld, the methodology is transferable to other use cases. Moreover, the BSDF parameters can be reparameterized to account for different materials and their specific optical properties, even in the absence of real images. Future work has to investigate whether the classifier can be trained on specific error classes rather than in a binary setup, enabling a more detailed analysis of individual defect types. As Xin et al. [14] found, it can sometimes be beneficial to perform inspections on 3D measurements of welds. Therefore, it is necessary to investigate whether synthetic 3D data can also be used for training neural networks.

Furthermore, the results show that object detectors trained on synthetic data already achieve promising performance, although their test metrics are not yet sufficient for direct deployment in production environments. A potential solution to this issue is to fine-tune the synthetically pretrained networks using real data, as proposed by Too et al. and Yosinski et al. [38,39]. Still, these results represent a significant step forward in enabling the use of neural networks prior to the actual start of production. However, one critical challenge remains: the evaluation of pretrained models in the absence of real validation data. This lack of ground-truth data poses a substantial obstacle to fully leveraging pretrained models before production begins. Addressing this challenge requires novel approaches to model validation and performance estimation under uncertainty. Potential strategies may include domain adaptation techniques, synthetic-to-real transfer evaluation, uncertainty quantification, and limited real data sampling from pilot production runs. Overcoming this hurdle is essential for the reliable deployment of pretrained neural networks in industrial applications where access to labeled real data is limited or delayed.

## Figures and Tables

**Figure 1 sensors-25-07301-f001:**
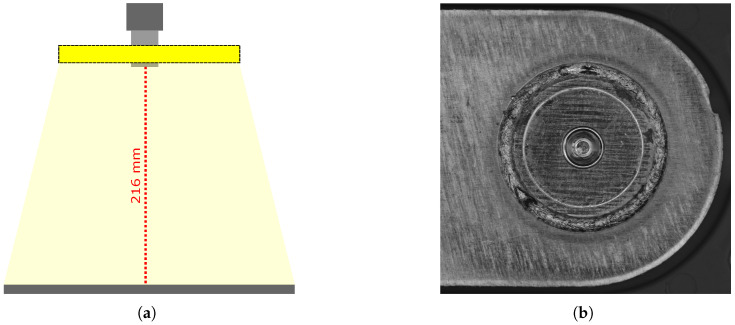
(**a**) Real image recording setup. A Basler a2A4200-12gmPRO camera is equipped with an Edmund Optics 56-872 lens with a focal length of 35 mm leading to a spatial resolution of approximately 13 μm. The scene is illuminated with an area emitter. (**b**) Real image of a circular weld.

**Figure 2 sensors-25-07301-f002:**
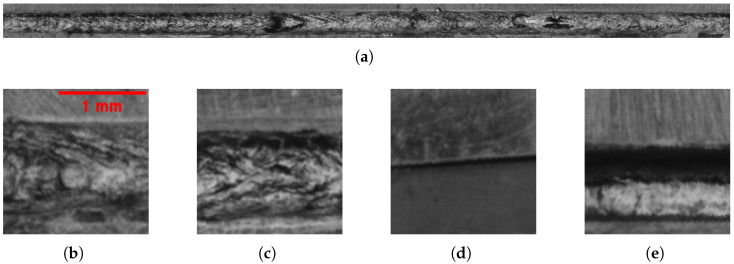
(**a**) Unwarped weld with dimensions of 35 mm × 1.66 mm. (**b**) OK weld. (**c**) OK weld with soot at the top edge of the weld. (**d**) NOK missing weld. (**e**) NOK weld with long hole defect.

**Figure 3 sensors-25-07301-f003:**
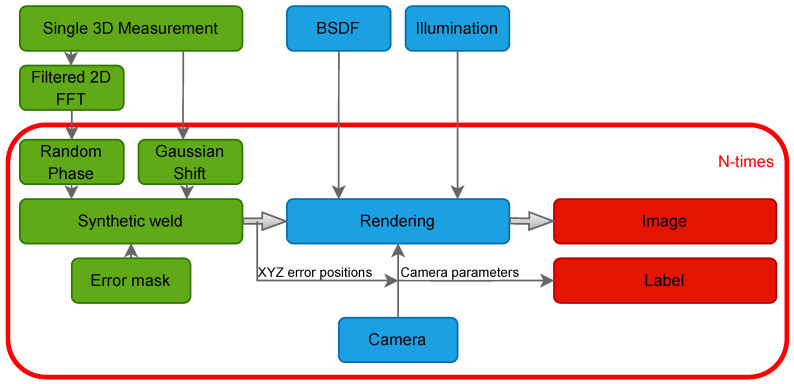
Scheme of dataset generation of synthetic welds from single 3D measurement to rendered images and labels.

**Figure 4 sensors-25-07301-f004:**
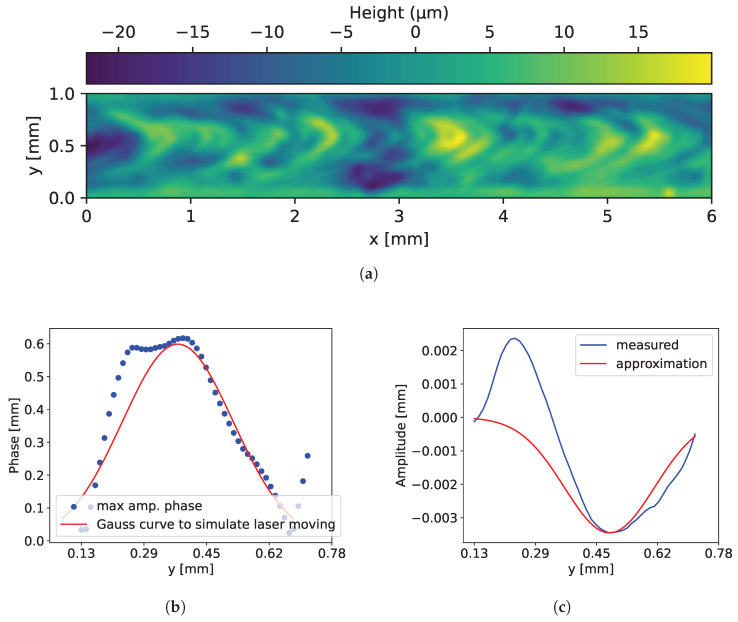
(**a**) White light interferometer topography measurement of a weld. (**b**) Phases of dominant amplitudes as function of y position. (**c**) Mean of the heights as function of y-position.

**Figure 5 sensors-25-07301-f005:**
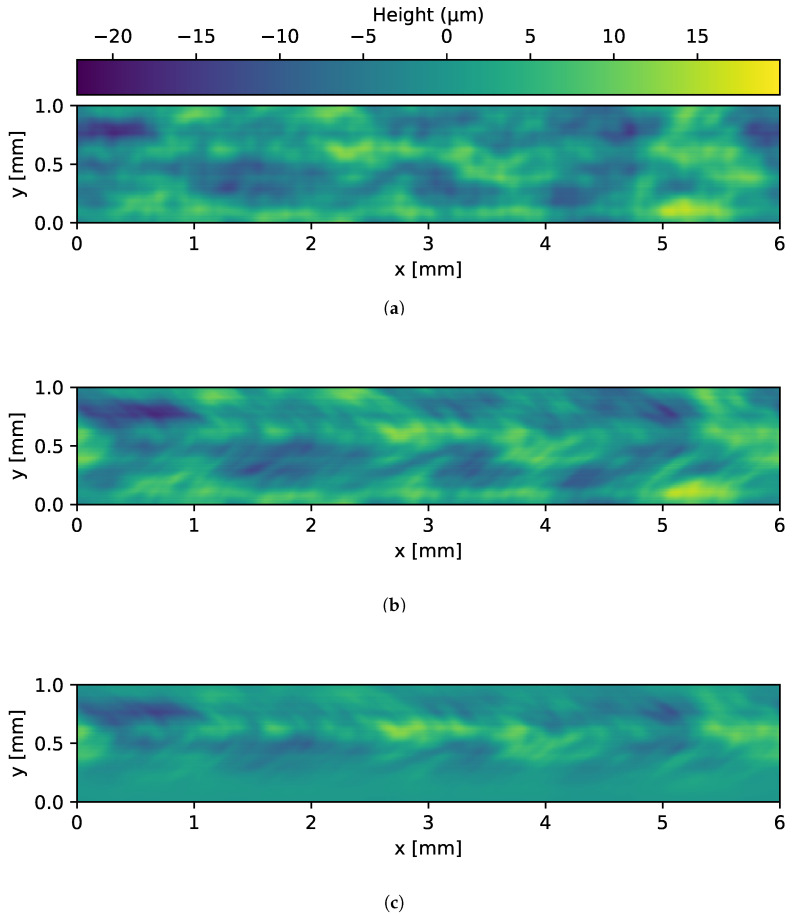
(**a**) Example of a synthetic surface obtained by the procedure described in Equation (Equation 2). (**b**) Same surface as shown in (**a**), but with an additional row-wise Gaussian pixel shift in the x-direction. (**c**) Same surface as shown in (**b**), but with additionally reduced weld edge heights.

**Figure 6 sensors-25-07301-f006:**
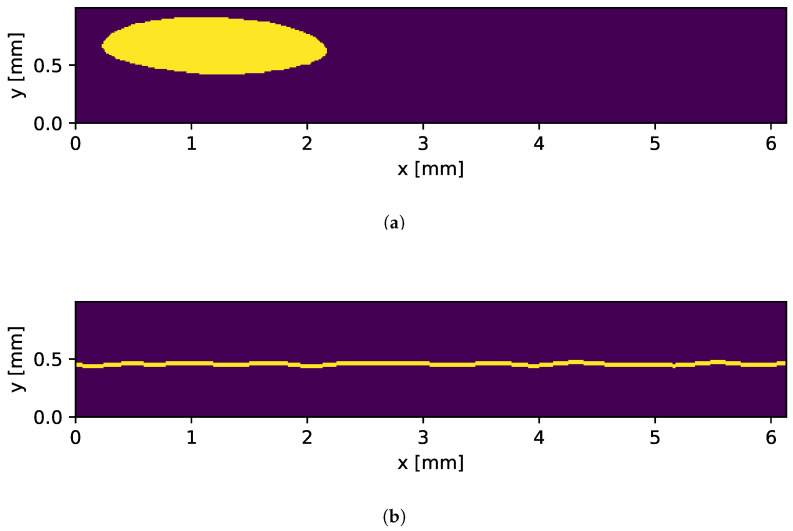
The yellow areas are removed from the mesh. (**a**) Hole mask. (**b**) Missing weld mask.

**Figure 7 sensors-25-07301-f007:**
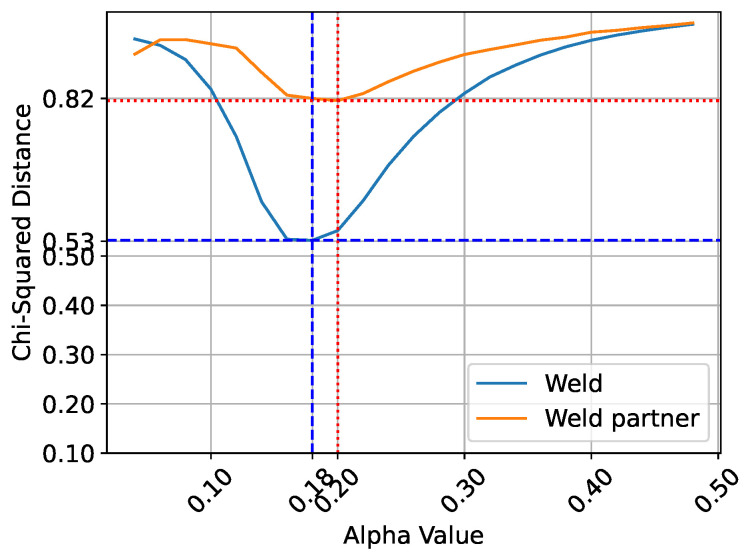
Distribution of the Chi-square distance as a function of the alpha values of the weld and weld partner. The blue line indicates the minimum Chi-square distance for the weld at an alpha value of 0.18, while the orange line shows the minimum distance for the weld partner at an alpha value of 0.20.

**Figure 8 sensors-25-07301-f008:**
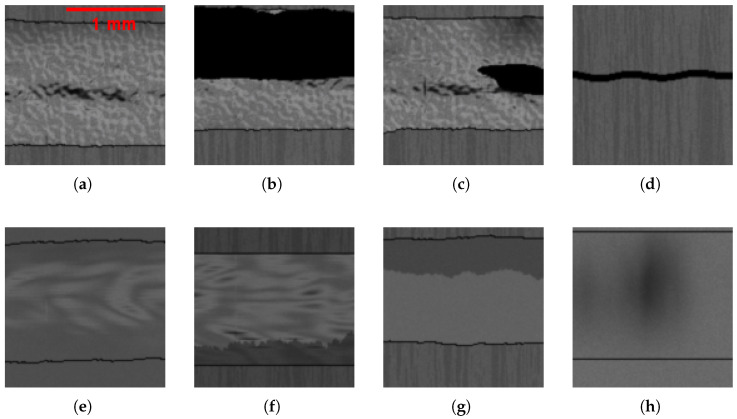
Example of synthetic weld images. (**a**) OK weld with best model performance. (**b**) NOK weld with long hole defect. (**c**) NOK weld with small hole defect. (**d**) NOK weld with missing weld. (**e**) Weld with random border, no Perlin noise, reduced edge height. (**f**) Weld without random border, no edge height reduction, with soot. (**g**) No weld topography, with soot and surface variance. (**h**) No weld topography, no random border, with soot.

**Figure 9 sensors-25-07301-f009:**
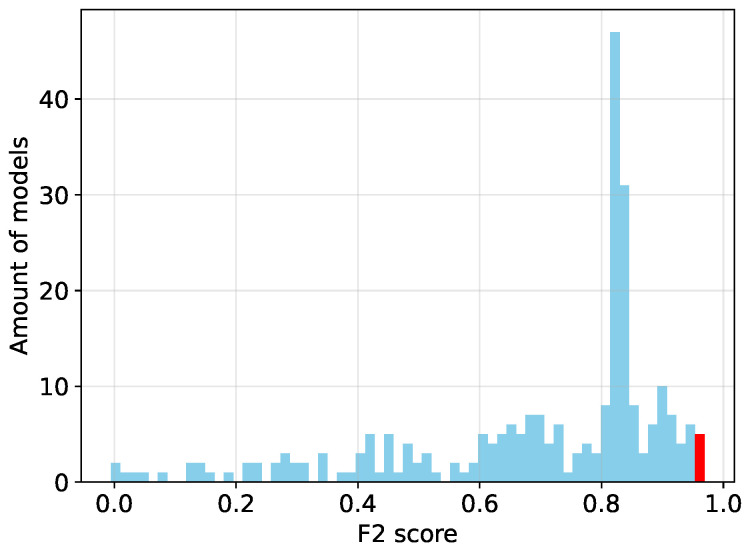
Distribution of $F_{2}$ scores of all 256 trained models. The best model of the red bar is further analyzed in Section 8.3.

**Figure 10 sensors-25-07301-f010:**
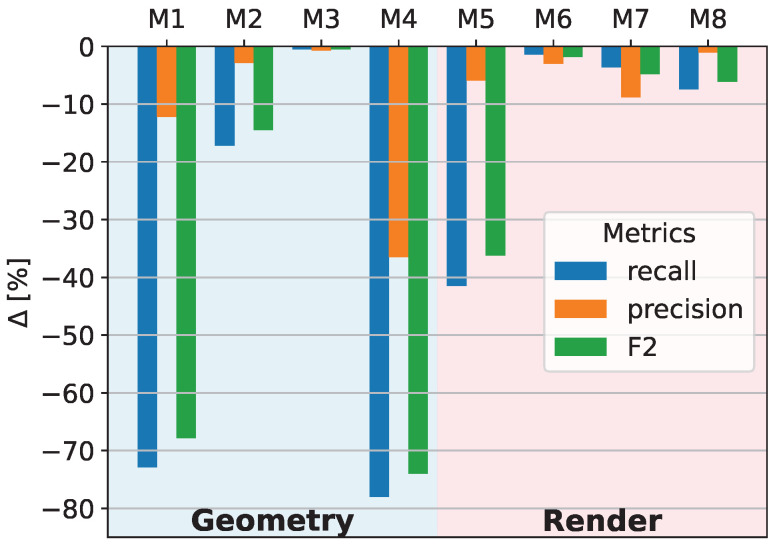
Sensitivity analysis of the test metrics to the individual parameters. In each model, one of the parameters is switched in comparison to the best model; see Table 2.

**Table 1 sensors-25-07301-t001:** Metrics of the best models chosen on $F_{2}$ score. Where Model 1 and 3 are trained on synthetic data and Models 2 and 4 on real data. All models are tested on real image data.

#	Model Name	Train	P	R	$F_{2}$
1	YOLO NAS S	Synth	0.487	0.833	0.729
2	YOLO NAS S	Real	0.955	0.958	0.956
3	YOLOv11 Class.	Synth	0.939	0.987	0.977
4	YOLOv11 Class.	Real	1.0	1.0	1.0

**Table 3 sensors-25-07301-t003:** (1) Best dataset, all parameters are enabled (•) except for modeled soot at edges (◦). (2) Only the three most influential parameters enabled, identified in Figure 10. (3) Only weld topography and illumination enabled. (4) Only Perlin noise and illumination enabled.

	Weld Topography	Reduced Weld Edges	Random Contour	Perlin Noise	Modeled Illumination	Partner Variances	Soot on Weld	Soot at Edges	R	P	*F* _2_
(1) best model	•	•	•	•	•	•	•	◦	0.987	0.939	**0.977**
(2) core model	•	◦	◦	•	•	◦	◦	◦	0.976	0.976	**0.976**
(3) topography model	•	◦	◦	◦	•	◦	◦	◦	0.557	0.943	0.607
(4) Perlin model	◦	◦	◦	•	•	◦	◦	◦	0.898	0.561	0.802

## Data Availability

The synthetic data supporting the conclusions of this article will be made available by the authors on request. Restrictions apply to the availability of the real image data. Data were obtained from Swoboda Wiggensbach KG and are available from the author Stefan Maier with the permission of Swoboda Wiggensbach KG.

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
