# Peer review of "Synthetic Data Generation for AI-Based Quality Inspection of Laser Welds in Lithium-Ion Batteries"

_sensors, 2025, doi:10.3390/s25237301_

Round 1
Reviewer 1 Report
Comments and Suggestions for Authors
This paper addresses the AI-based visual inspection of laser weld beads, presenting a methodology for training neural networks using synthetically generated data via Physically Based Rendering (PBR). The paper is well-structured, with a solid experimental design and valuable results, especially in the context of the growing need for automated quality control where real training data is limited. The innovative approach of utilizing synthetic data for pre-training, while also analyzing key modeling parameters, is highly practical and promising for industrial applications.
Specific Comments:
- Introduction Section: It is recommended to enhance the background on laser welding applications by including descriptions of its use in multiple industries, such as power battery manufacturing, automotive electronics, aerospace, and consumer electronics. Highlighting the demand for weld inspection across these fields will better emphasize the significance of the study and its broader applicability. Please consider referencing the following literature as support: Int. J. Extrem. Manuf. 7 (2025) 032001.
- The explanation of the weld bead geometry and lighting modeling is somewhat complex. It would be helpful to include a conceptual diagram illustrating the workflow from real data collection, PBR parameter setting, rendering generation, automated labeling. This would help readers quickly understand the method's logic.
- It would be beneficial to include statistics on training time and computational resources required for different parameter combinations, helping to demonstrate the practical feasibility of this approach for industrial deployment.
- Although the paper conducts a sensitivity analysis on key parameters, it does not discuss the model's applicability to other weld structures or materials (e.g., aluminum, copper alloys). I recommend adding a section in the "Outlook" to address cross-domain transferability and potential challenges when applying this method to other industrial contexts.
- Overall, the language is clear, but some sections contain long sentences. Breaking them into shorter sentences could improve readability and comprehension.
Author Response
Dear Reviewer,
Please see the attachment. Thank you for reviewing.
Sincerely,
The Authors

Reviewer 2 Report
Comments and Suggestions for Authors
Comments and Suggestions for Authors
In this paper, the authors provide a study to describe a methodology for training AI models using synthetically generated image data. The synthetic images are created using Physically Based Rendering, which enables precise control over rendering parameters and facilitates automated labeling. This approach allows for a systematic analysis of parameter importance and bypasses the need for large real training datasets. As a case study, used in this paper focuses on the inspection of laser weld beads in battery connectors for fully electric vehicles—a particularly demanding application due to the criticality of each weld. The results demonstrate the effectiveness of synthetic data in training robust AI models, thereby providing a scalable and efficient alternative to traditional data acquisition and labeling methods.
Some comments and suggestions for authors could be considered.
- In introduction section the authors provide a limited review of the state of the art regarding references of AI-Based quality inspection of laser weld beads. Relevant recent references are missing that must be included in introduction section.
- In introduction section the authors provide a combination of introduction of the state of the art reported in literature with methodology of this work that must be described in materials and methos section. Move section 3. Real image recording setup including Fig. 1 and Fig. 2 to 5. Methods section.
- AI-Based quality inspection of laser weld beads has been widely reported in recent years. In this sense, the manuscript does not clearly identify how its approach significantly advances the state of the art. In this sense, this document does not clearly describe the methodology of mathematical models and parameters used for inspection in the introduction section.
- Caption of Fig. 5 is too long. Provide a synthetic caption and describe each Figure in the main text.
- The results are based on the comparison of numerical results of the complex surface structures and high variability in appearance analyzed. Move Fig. 5 to results section to describe the welded defects are related to the numerical results models.
- The conclusions section should include significant quantitative results of the study and not only qualitative results.
Author Response

(The authors gave the same response as above.)

Reviewer 3 Report
Comments and Suggestions for Authors
- The title is lengthy and requires condensation, with the type of batteries used (e.g., lithium-ion batteries) specified for greater practical clarity.
- The abstract includes a repeated sentence that already appears in the introduction (page 1, line 25). This repetition should be removed.
- The use of the first-person plural pronoun we is not appropriate in formal academic writing.
- It is recommended to replace the term beads with joints, as the latter is more precise in the context of battery welding.
- The research problem is not clearly articulated. The introduction focuses on data limitations without clarifying the specific scientific gap or the study’s contribution beyond previous work.
- References [12], [13], and [14] are not directly related to laser welding of batteries and should be distinguished from references [10] and [11], which are more relevant to the targeted industrial application.
- The caption of Figure 1 is imprecise and does not adequately describe the optical setup and imaging configuration.
- The source of Equations (1–8) is not cited. It should be clarified whether these equations were developed by the authors or derived from previous studies.
- The number of real samples (50) is very limited, which weakens the statistical robustness and reduces the representativeness of real industrial conditions.
10 The justification for considering the total number of images (5980) sufficient for neural network training is not provided, particularly regarding statistical adequacy and performance stability.
11 . Line 142 states that the modeling is visually rather than physically driven, which reduces the physical realism of the model.
12. The selection of the 2D Fourier Transform as the primary surface generation method is
not adequately justified, especially considering more recent alternatives such as Neural Style Transfer or GAN-based generation.
13.The potential error rate in the automated labeling process is not addressed, despite its direct impact on training data reliability.
14. The explanation of Figure 8, which presents the distribution of F2 scores across the 256 trained models, is insufficient and does not clarify the causes of performance variation.
15. The physical nature of the defects corresponding to the hole and missing bead categories is not clearly defined in terms of metallurgical or structural discontinuities, which limits the engineering relevance of the findings.
Author Response

(The authors gave the same response as above.)

Reviewer 4 Report
Comments and Suggestions for Authors
The manuscript proposes a systematic methodology for developing AI-based weld bead inspection systems trained exclusively on synthetically generated data using Physically Based Rendering (PBR). The case study on laser weld bead inspection in battery connectors for electric vehicles demonstrates strong applicability and industrial relevance. The work appears comprehensive; however the manuscript would benefit from improved organization, more focused discussion of mechanisms, clearer quantitative reporting, and enhanced presentation quality.
- The authors are required to clearly articulate the main research objectives and hypotheses in the Introduction section. Currently, they are embedded within the narrative rather than stated explicitly.
- The paper would benefit from a comparative discussion of model performance on synthetic vs. real data. The authors are required to provide quantitative metrics that demonstrate domain transfer effectiveness.
- While numerous parameters (geometry, illumination, BSDF, Perlin noise) are modeled, the authors are required to explain the rationale for selecting parameter ranges and how these affect model generalization.
- Minor grammatical issues are present. Sentences such as “Therefor we investigate…” should be corrected for academic clarity. Consistent use of technical terminology (e.g., rendering parameters vs. illumination effects) is recommended.
Author Response

(The authors gave the same response as above.)

Round 2
Reviewer 2 Report
Comments and Suggestions for Authors
The authors have made and corrected the comments and suggestions in the review. Its publication in its present form is recommended.